# Individual responses to a single oral dose of albendazole indicate reduced efficacy against soil-transmitted helminths in an area with high drug pressure

**Martin Walker**[1,2]*, **Piet Cools**[3], **Marco Albonico**[4,5], **Shaali M. Ame**[6], **Mio Ayana**[7], **Daniel Dana**[7], **Jennifer Keiser**[8,9], **Leonardo F. Matoso**[10,11], **Antonio Montresor**[12], **Zeleke Mekonnen**[7], **Rodrigo Corrêa-Oliveira**[10], **Simone A, Pinto**[10], **Somphou Sayasone**[13], **Jozef Vercruysse**[3], **Johnny Vlaminck**[3], **Bruno Levecke**[3]*

1 Department of Pathobiology and Population Sciences, Royal Veterinary College, Hatfield, United Kingdom, 2 London Centre for Neglected Tropical Disease Research, Imperial College London, London, United Kingdom, 3 Department of Virology, Parasitology and Immunology, Ghent University, Merelbeke, Belgium, 4 Center for Tropical Diseases, Sacro Cuore Don Calabria Hospital, Negrar, Italy, 5 Department of Life Sciences and Systems Biology, University of Turin, Turin, Italy, 6 Laboratory Division, Public Health Laboratory-Ivo de Carneri, Chake Chake, United Republic of Tanzania, 7 Jimma University Institute of Health, Jimma University, Jimma, Ethiopia, 8 Department of Medical Parasitology and Infection Biology, Swiss Tropical and Public Health Institute, Basel, Switzerland, 9 University of Basel, Basel, Switzerland, 10 Laboratory of Molecular and Cellular Immunology, Research Center René Rachou—FIOCRUZ, Belo Horizonte, Brazil, 11 Nursing school, Federal University of Minas Gerais, Brazil, 12 Department of Control of Neglected Tropical Diseases, World Health Organization, Geneva, Switzerland, 13 Lao Tropical and Public Health Institute, Ministry of Health, Vientiane, Lao People's Democratic Republic

* mwalker@rvc.ac.uk (MW); bruno.levecke@ugent.be (BL)

**Data Availability Statement:** The data analyzed in this manuscript are publicly available https://doi.org/10.1371/journal.pntd.0007471.s004.

## Abstract

### Background

Albendazole (ALB) is administered annually to millions of children through global deworming programs targeting soil-transmitted helminths (STHs: *Ascaris lumbricoides*, *Trichuris trichiura* and hookworms, *Necator americanus* and *Ancylostoma duodenale*). However, due to the lack of large individual patient datasets collected using standardized protocols and the application of population-based statistical methods, little is known about factors that may affect individual responses to treatment.

### Methodology/Principal findings

We re-analyzed 645 individual patient data from three standardized clinical trials designed to assess the efficacy of a single 400 mg oral dose of ALB against STHs in schoolchildren from different study sites, each with varying history of drug pressure based on duration of mass drug administration programs: Ethiopia, low; Lao People's Democratic Republic (PDR), moderate; Pemba Island (Tanzania), high. Using a Bayesian statistical modelling approach to estimate individual responses (individual egg reduction rates, $ERR_i$), we found that efficacy was lower in Pemba Island, particularly for *T. trichiura*. For this STH, the proportion of participants with a satisfactory response ($ERR_i \geq 50\%$), was 65% in Ethiopia, 61% in

**Funding:** MW acknowledges funding from UK Research and Innovation via Research England's Connecting Capabilities Fund (Bloomsbury SET, https://bloomsburyset.org.uk; BSA34) and from a Wellcome Biomedical Resources Grant (https://wellcome.org; 208378/Z/17/Z,) with the Infectious Diseases Data Observatory (www.iddo.org). The Starworms study was supported by a grant from the Bill and Melinda Gates foundation (https://www.gatesfoundation.org, OPP1120972, PI is BL, www.starworms.org). PC was financially supported by a grant from the Bill and Melinda Gates foundation (OPP1120972, www.starworms.org). JV was financially supported through an International Coordination Action of the Flemish Research Foundation. The funders had no role in study design, data collection and analysis, decision to publish, or preparation of the manuscript.

**Competing interests:** The authors have declared that no competing interests exist.

Lao PDR but only 29% in Pemba Island. There was a significant correlation between $ERR_i$ and infection intensity prior to drug administration ($ERR_i$ decreasing as a function of increasing infection intensity). Individual age and sex also affected the drug response, but these were of negligible clinical significance and not consistent across STHs and study sites.

## Conclusions/Significance

We found decreased efficacy of ALB against all the STHs analyzed in Pemba Island (Tanzania), an area with high drug pressure. This does not indicate causality, as this association may also be partially explained by differences in infection intensity prior to drug administration. Notwithstanding, our results indicate that without alternative treatment regimens, program targets will not be achievable on Pemba Island because of inadequate efficacy of ALB.

## Trial registration

The study was registered on Clinicaltrials.gov (ID: NCT03465488) on March 7, 2018.

## Author summary

More than 500 million children worldwide receive a single oral dose of albendazole or mebendazole annually to reduce disease caused by intestinal worms (roundworm, whipworm and hookworm). However, it is unclear whether individuals respond differently to treatment. We re-analyzed 645 individual patient data from three standardized clinical trials with albendazole using a statistical method and explored how drug responses among individuals from study sites with different prior rounds of drug distribution. We found that individuals' responses were worse when drug pressure and infection intensity were greater. Individual age and sex also affected the response, but these were often of negligible clinical significance and not consistent across worm species and study sites. We confirmed that albendazole is ineffective against whipworm on Pemba Island, but given the high intensity of infection at this study side, it remains unclear whether this treatment failure is explained by high drug pressure (and emerging anthelmintic resistance) or high infection intensity. Nevertheless, our results indicate that alternative treatment regimens are required to control intestinal worms in Pemba Island.

## Introduction

Benzimidazole drugs are administered annually to more than 500 million children as part of deworming programs that aim to eliminate morbidity caused by soil-transmitted helminths (STHs; *Ascaris lumbricoides* or roundworm, *Trichuris trichiura* or whipworm and the hookworms, *Necator americanus* and *Ancylostoma duodenale*) [1–3]. Benzimidazoles inhibit the polymerization of tubulins into microtubules (key components for cell transport and energy metabolism), that in turn inhibit excretion of waste products and protective factors from worm cells, which ultimately leads to a progressive depletion of energy reserves [4,5]. The current benzimidazoles administered in large scale deworming programs are albendazole (ALB) and mebendazole (MEB), both given as a single oral dose of 400 mg and 500 mg, respectively [6]. At the population level, these two drugs show a very high therapeutic efficacy (measured as the percent reduction in arithmetic mean egg counts following drug administration in a population, the so-

called egg reduction rate, ERR) against *A. lumbricoides* but a poor efficacy against *T. trichiura*. For hookworm infections, ALB is significantly more potent than MEB [7,8].

At the individual level, it currently remains unclear what factors affect a patient's response to benzimidazoles, which consequently impedes further optimizing of treatment regimens (e.g. impact of dose on the drug response for the different STH infections [9,10]) and interpreting apparent treatment failures (e.g. low therapeutic efficacy when infection intensity is high [11,12] or when the follow-up period is not optimal [13]). The lack of insights into the individual patient response can be explained by both the scarcity of large individual patient datasets collected during standardized clinical trials that are shared and the relative complexity of individual-based analytical approaches [14–16] compared to their more accessible population-based statistical counterparts [17].

In this study, we analyze publicly available individual patient data from three standardized clinical trials that were designed to assess the efficacy of a single 400 mg oral dose of ALB against STH infections in schoolchildren [18–20]. The trials were conducted in three study sites in Ethiopia, Lao People's Democratic Republic (PDR) and Pemba Island (Tanzania), each with a varying history of drug pressure (i.e., different number of past rounds of mass drug administration, MDA). For each of the three STHs, we used an individual-based Bayesian modelling approach [15,16,21] to explore how study site (drug pressure history), age and sex, follow up time, co-infection and intensity of infection affect drug efficacy.

## Methods

### Ethics statement

The study protocol has been reviewed and approved by the Ethics Committee of the Faculty of Medicine and Health Sciences of Ghent University, Belgium (Ref. No B670201627755; 2016/0266). The trial protocol was subsequently reviewed and approved by responsible bodies at each study site (Ethical Review Board of Jimma University, Jimma, Ethiopia: RPGC/547/2016; National Ethics Committee for Health Research, Vientiane, Lao PDR: 018/NECHR; Zanzibar Medical Research and Ethics Committee, Tanzania: ZAMREC/0002/February/2015). The study was retrospectively registered on Clinicaltrials.gov (ID: NCT03465488) on March 7, 2018. Parent (s)/guardians of participants signed an informed consent form indicating that they understood the purpose and procedures of the study, and that they allowed their child to participate. If the child was ≥5 years, he or she had to orally assent to participate. Participants of ≥12 years of age were only included if they signed an informed consent form indicating that they understood the purpose and the procedures of the study and were willing to participate.

### Study design and population

Details on the trial design have been presented elsewhere [20]. Briefly, the trials were designed to assess an equivalence in treatment efficacy of a single 400 mg oral dose of ALB against STH infections in schoolchildren measured by a variety of diagnostic methods in three study sites with different drug pressure history. In Ethiopia, large deworming programs using ALB began in 2015; in Lao PDR, using MEB, in 2007, and in Pemba Island (Tanzania), using ALB, in 1994. The trials were standardized by means of study population (schoolchildren age 6–14; same inclusion and exclusion criteria), sample size per STH (*A. lumbricoides*: 111–193; *T. trichiura*: 105–245; hookworm: 90–228), sampling effort (one stool sample per child both prior to and after drug administration), diagnostic methods (single and duplicate Kato-Katz thick smear, Mini-FLOTAC, FECPAK$^{G2}$ and qPCR), origin of drug (GlaxoSmithKline, batch number N˚ 335726), dose (single oral 400 mg) and follow-up period (8–34 days, with 87% between the recommended 2 and 3 weeks [17]).

## Statistical analysis

Although all stool samples were screened and infection diagnosed with different diagnostic methods (single and duplicate Kato-Katz thick smear, Mini-FLOTAC, FECPAK[G2] and qPCR), we restricted the statistical analysis of the present study to the duplicate Kato-Katz thick smear only. This is because: (i) Kato-Katz thick smear is the most commonly applied method for monitoring and evaluation activities in STH MDA programs; (ii) of all the applied methods in these trials, Kato-Katz thick smear is the only method that has been recommended so far by the World Health Organization (WHO) to assess therapeutic efficacy [17]; (iii) the other methods resulted in egg counts that were significantly different from those reported by Kato-Katz thick smear (Mini-FLOTAC and FECPAK[G2]) or were measured in different units (i.e. qPCR) [18, 22]. We calculated the population-based therapeutic efficacy, $ERR_P$, of a single 400 mg oral dose of ALB separately for each STH species in each country using the fecal egg count (FEC) data before and after drug administration (a FEC being the number of eggs counted from a single Kato-Katz slide, such that two FECs are measured from duplicate Kato-Katz slides from each individual at each time point) by applying **Eq (1)**. The 95% confidence intervals (CIs) around $ERR_P$ were calculated using a non-parametric block bootstrap approach [23].

$$ERR_P = 100 \times \left( 1 - \frac{\text{arithmetic mean fecal egg count after drug administration}}{\text{arithmetic mean fecal egg count before drug administration}} \right) \quad (1)$$

We used our previously developed Bayesian modelling framework [15,16,21] to estimate individual ERRs, denoted $ERR_i$, from the FECs before and after drug administration. Full mathematical details are given in **S1 Text**. Briefly, we used a negative binomial mixed (fixed and random effects) regression structure to model a multiplicative change in FECs from before to after drug administration, such that $ERR_i$ is given by an expression of the general form $1 - \exp(\boldsymbol{\beta}\mathbf{X}_i + b_{1i})$. Here, $\mathbf{X}_i$ and $\boldsymbol{\beta}$ denote interaction terms and associated regression coefficients describing the effect of fixed effects covariates (age, sex, follow-up time, co-infection status and study site) on the response to treatment (i.e., FECs measured after drug administration) and $b_{1i}$ is a random effect 'gradient' term allowing the response to vary among individuals (sharing covariates). We also included random effects terms to account for correlation among FECs measured from the same individual (duplicate Kato-Katz is performed on the same sample) and from the same school (i.e., random intercept terms for individual and school). The covariance between individual intercept and 'gradient' terms quantifies the association between an individual's FEC before drug administration and their response following administration of ALB. This allowed us to estimate the effect of infection intensity on individual ERRs. We fitted models separately to the data from each STH species (*A. lumbricoides*, *T. trichiura* and hookworm), combining FEC data from different countries to estimate study site (with different drug pressures) effects on ERRs. The models were fitted using Bayesian Markov chain Monte Carlo sampling techniques implemented in Stan [24] called from R (version 3.6.3) [25] using the brms [26,27] package.

## Results

### Demographics and STH status of the complete cases

Complete data (i.e., participants who provided a stool sample before and after drug administration which was analyzed by duplicate Kato-Katz thick smear, Mini-FLOTAC, FECPAK[G2] and qPCR) were available for 645 children across the three study sites (Ethiopia: 161 cases; Lao PDR: 239 cases; Pemba Island: 245 cases). The demographics of the complete cases is described

in detail elsewhere. Briefly, the overall sex ratio was 1:1.1 (310 males *vs.* 335 females). The median age (interquartile range) of the children was 11.0 years (9.0–12.0). In total, there were 441 complete cases (i.e. positive by any diagnostic and followed up after treatment) for *A. lumbricoides* (Ethiopia: 137, Lao PDR: 111, Pemba Island: 193), 456 for *T. trichiura* (Ethiopia: 106, Lao PDR: 105, Pemba Island: 245) and 457 for hookworm (Ethiopia: 90, Lao PDR: 228, Pemba Island: 139). The qPCR results showed that all individuals who excreted hookworm eggs were infected with *N. americanus*, eight of whom also had *Ancylostoma* DNA (these coinfected individuals were from one school on Pemba Island; see Vlaminck et al. [19] for discussion on the potential differential susceptibility of *N. americanus* and *Ancylostoma* species to ALB). The fraction of cases that were positive by duplicate Kato-Katz thick smear at baseline was 366/441 (83%) for *A. lumbricoides* (Ethiopia, 121/137, 88%; Lao PDR, 97/111, 87%; Pemba Island, 148/193, 77%), 423/456 (93%) for *T. trichiura* (Ethiopia, 86/106, 81%; Lao PDR, 92/105, 88%; Pemba Island, 245/245, 100%) and 374/457 (82%) for hookworm (Ethiopia, 67/90, 74%; Lao PDR, 215/228, 94%; Pemba Island: 92/139, 66%). The total egg counts for each individual at both the baseline and follow-up are shown in **Fig 1** for the different STHs and study sites separately.

## Population-based therapeutic drug efficacy

The population-based therapeutic efficacy of ALB, expressed as $ERR_P$, and measured by duplicate Kato-Katz thick smear is reported in **Table 1**. Based on the classification criteria recommended by the WHO [17], the efficacy of ALB against *A. lumbricoides* can be judged as 'satisfactory' ($ERR_P \geq 95\%$) in all three countries, and against hookworm ($ERR_P \geq 90\%$) in Ethiopia and Lao PDR. The efficacy of ALB against hookworm in Pemba Island is 'doubtful' ($80\% \leq ERR_P < 90\%$) and against *T. trichiura*, it is 'reduced' in Pemba Island ($ERR_P < 40\%$) and 'doubtful' ($40\% \leq ERR_P < 50\%$) in both Ethiopia and Lao PDR.

## Individual-based therapeutic drug efficacy

Individual-level estimates of the ERR, denoted $ERR_i$, for each of the three STH infections (*A. lumbricoides*, *T. trichiura* and hookworm) in each of the three sites (Ethiopia, Lao PDR and Pemba Island, Tanzania) are shown in **Fig 2**. For *T. trichiura*, the proportion of participants with a satisfactory drug response ($ERR_i \geq 50\%$) was 65% in Ethiopia, 61% in Lao PDR and 29% in Pemba Island. For hookworms, the difference in a satisfactory response ($ERR_i \geq 90\%$) across the different study sites was less pronounced, ranging from 86% in Ethiopia to 63% in Pemba Island. For *A. lumbricoides*, a high proportion (>95%) of participants showed a satisfactory response ($ERR_i \geq 95\%$). See **S1 Table** for the estimates of the proportions of individuals with a 'satisfactory' response by infection in each study site.

Males and those aged 13–14 infected with *A. lumbricoides* had a statistically better response following administration of ALB, and individuals in Lao PDR and Pemba Island a worse response (**Fig 3A** and **S2 Table**). However, the clinical relevance of these associations is negligible (**Fig 4A**) due to the generally very high $ERR_i$ among all participants infected with *A. lumbricoides* (**Fig 2A**). Similarly, although a statistically significant worse response to ALB was associated with children on Pemba Island infected with hookworm (**Fig 3C** and **S4 Table**), this had a relatively limited impact on $ERR_i$ (**Fig 4C**). By stark contrast, responses to treatment among children infected with *T. trichiura* on Pemba Island were significantly (**Fig 3B** and **S3 Table**) and substantively (**Fig 4B**) worse in Ethiopia and Lao PDR. Indeed, a 'typical' child in the Pemba Island study had an $ERR_i$ that was not statistically discernable from 0%, indicating complete treatment failure (**Fig 4B**). For all infections, a negative correlation was identified

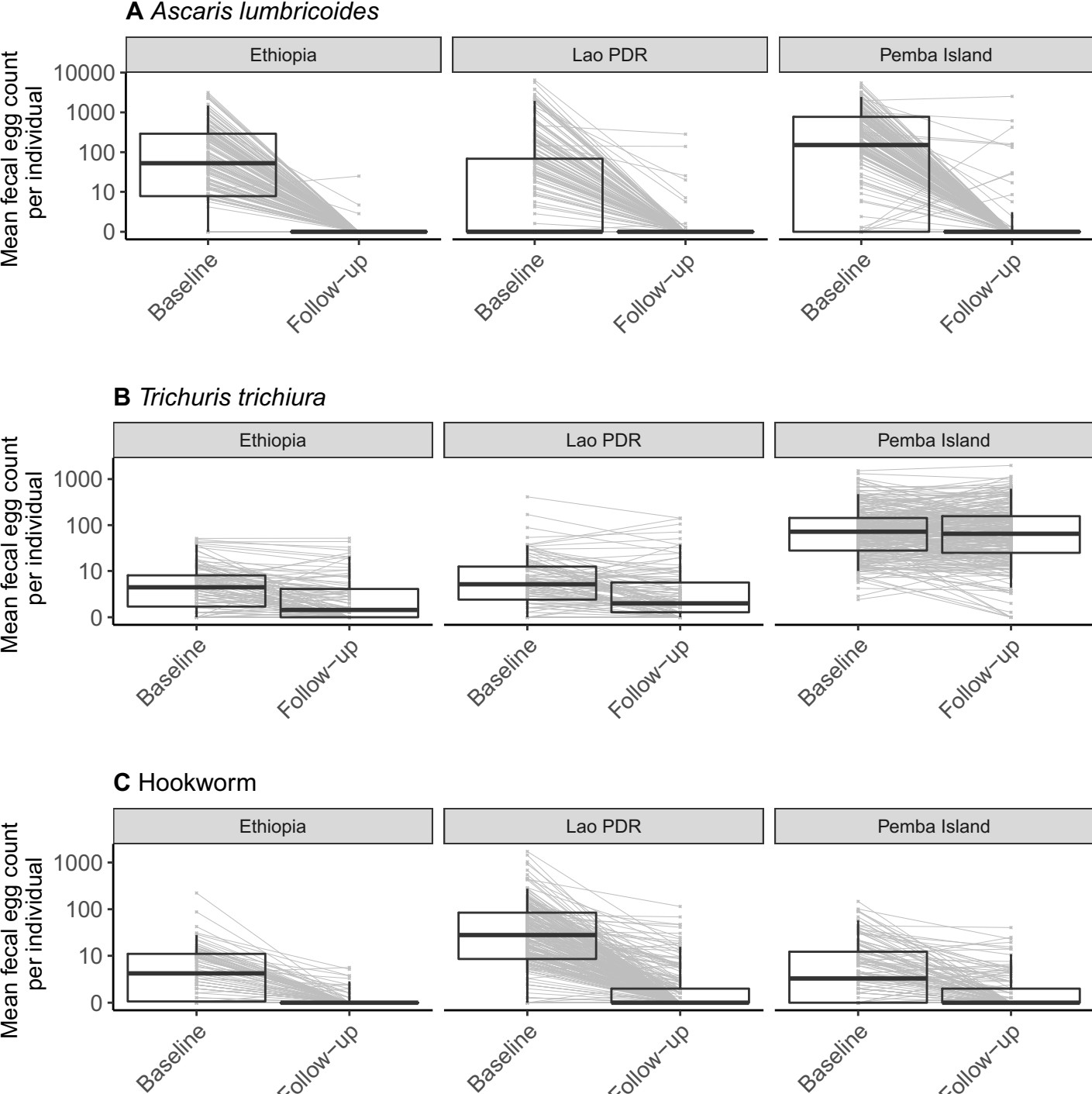

**Fig 1. Mean fecal egg count per individual before and after administration of a single oral dose albendazole.** The mean number of eggs counted across duplicate Kato-Katz slides from one stool sample for each individual (mean fecal egg count) before (baseline) and after administration of a single 400 mg oral dose of albendazole (follow-up) for *Ascaris lumbricoides* (**panel A**), *Trichuris trichiura* (**panel B**) and hookworm (**panel C**) in Ethiopia, Lao PDR and Pemba Island (Tanzania). The boxplots show the median, interquartile range, 5[th] and 95[th] percentiles of the data. Grey lines join mean fecal egg counts measured from the same individual.

**Table 1. Population-based therapeutic efficacy of albendazole against soil-transmitted helminth infections in three study sites.** The population-based therapeutic efficacy is expressed as the egg reduction rate, $ERR_P$, defined as the reduction in the arithmetic mean fecal egg count (FEC) following administration of a single 400 mg oral dose of albendazole (see **Eq (1)**). The FECs were based on duplicate Kato-Katz thick smear. $N_{tot}$ represents the total number of participants screened for STH infection; $N_+$ the number of participants identified as infected at baseline by any of the four diagnostics (duplicate Kato-Katz thick smear, Mini-FLOTAC, FECPAK$^{G2}$ and qPCR) and, $N_{KK+}$ the number of participants infected at baseline by duplicate Kato-Katz thick smear. The 95% confidence intervals (CIs) were calculated using a non-parametric block bootstrap approach [23].

| Study site | *Ascaris lumbricoides* | | *Trichuris trichiura* | | Hookworm | |
|---|---|---|---|---|---|---|
| | $N_{KK+}/N_+$ | $ERR_P$ (%) (95% CI) | $N_{KK+}/N_+$ | $ERR_P$ (%) (95% CI) | $N_{KK+}/N_+$ | $ERR_P$ (%) (95% CI) |
| Ethiopia ($N_{tot}$ = 161) | 121/137 | 99.9 (99.8;100) | 86/106 | 48.1 (33.8; 63.4) | 67/90 | 96.3 (92.1; 98.6) |
| Lao PDR ($N_{tot}$ = 239) | 97/111 | 99.2 (97.9; 99.9) | 92/105 | 40.5 (4.6; 40.0) | 215/228 | 96.2 (93.9; 97.8) |
| Pemba Island ($N_{tot}$ = 245) | 148/193 | 96.7 (92.1; 99.6) | 245/245 | -4.9 (-22.3; 12.3) | 92/139 | 83.6 (74.2; 90.5) |

between an individual's FEC before drug administration and their subsequent response to ALB (**Fig 5**). That is, individuals with higher FECs tended to have lower $ERR_i$.

## Discussion

Benzimidazoles are widely administered as part of large-scale deworming programs to reduce STH-attributable morbidity in children. Yet little is known about factors that may affect individual responses following a single oral dose of benzimidazole drugs. In the present study, we analyzed using a Bayesian statistical approach [15,16,21], publicly available individual patient data from three standardized clinical trials that were designed to assess the efficacy of a single 400 mg oral dose of ALB against STH infections in schoolchildren [18–20]. The trials were conducted in three study sites (Ethiopia, Lao PDR and Pemba Island (Tanzania)), each with a different history of drug pressure (i.e., a different number of prior rounds of drug administration).

We found that the efficacy of ALB against *A. lumbricoides* (roundworm) infection was high across all study sites, with no measured covariates having a substantive effect on the response. Responses in participants infected with hookworm were also generally good, albeit with evidence that the efficacy of ALB in Pemba Island (Tanzania) was marginally lower than in either Ethiopia or Lao PDR. The starkest finding is that the treatment of children infected with *T. trichiura* (whipworm) in Pemba Island was ineffective, with estimates of the ERR not discernable from 0% at the population level and with only 29% of individuals having a nominally 'satisfactory' response ($ERR_i \geq 50\%$). For a 'typical' child in Pemba Island, the estimated $ERR_i$ was not different from 0%, indicating treatment failure. This is of major concern to the long-term effectiveness of the program.

Benzimidazoles are known to have poorer efficacy against whipworm than other STH species [7]; an $ERR_P \geq 50\%$ is deemed 'satisfactory', as opposed to $\geq 90\%$ and $\geq 95\%$ for hookworm and roundworm respectively [17]. On this basis, responses to ALB in the Ethiopian and Lao PDR study sites, although at a population-level would be deemed 'doubtful', were 'satisfactory' at an individual-level for a majority (>50%) of participants. By contrast, on Pemba Island, the population-level ERR was not statistically different from 0% and only 29% of participants had a satisfactory response. The poor efficacy of ALB in Pemba Island was reported in a previous analysis of these data [19] and three previous clinical trials [28–30]. Moreover, a recent global meta-analysis of randomized control trials found that the efficacy of ALB against *T. trichiura* has declined between 1995 and 2015 [7].

## A *Ascaris lumbricoides*

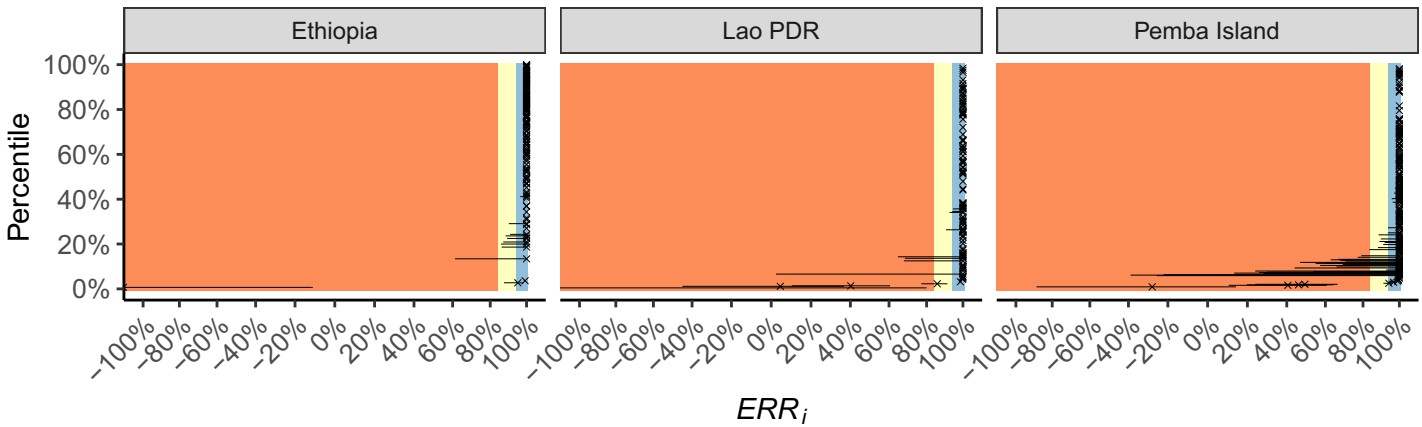

## B *Trichuris trichiura*

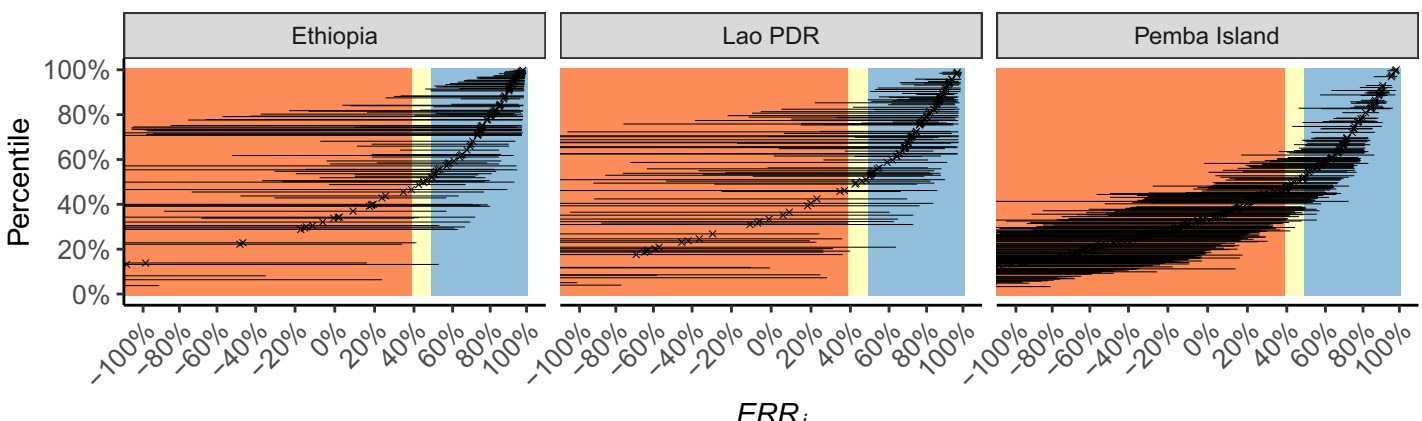

## C Hookworm

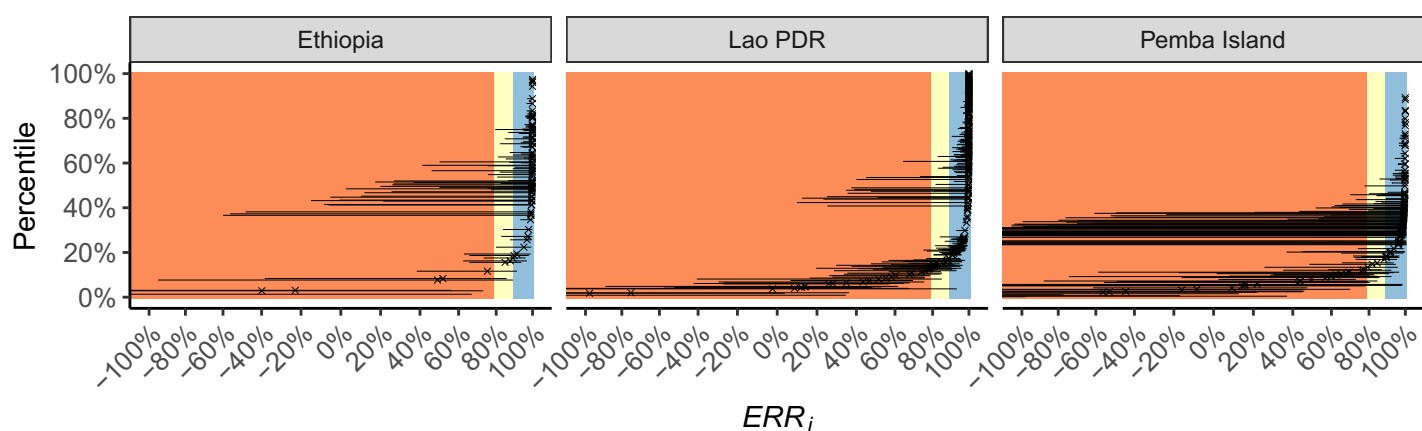

**Fig 2. Individual therapeutic efficacy of albendazole against soil-transmitted helminth infections estimated in three study sites.** The individual-based therapeutic efficacy is expressed as the reduction in fecal egg counts (FECs) following drug administration (egg reduction rate, $ERR_i$) and was estimated by fitting the Bayesian model [15,16,21] to FEC data on *Ascaris lumbricoides* (**panel A**), *Trichuris trichiura* (**panel B**) and hookworm (**panel C**) collected before and after administration of a single 400 mg oral dose of albendazole in three study sites, Ethiopia, Lao PDR and Pemba Island (Tanzania). Each cross denotes the posterior median of $ERR_i$ and horizontal lines denote 95% credible intervals. The estimates are ordered according to the percentile of the estimated median $ERR_i$. The areas shaded in orange, yellow and blue denote, respectively, the World Health Organization categories for 'reduced', 'doubtful' and 'satisfactory' population-based therapeutic efficacy ($ERR_P$, which differ for each infection). Negative estimates of $ERR_i$ correspond to FECs that increased after administration of albendazole.

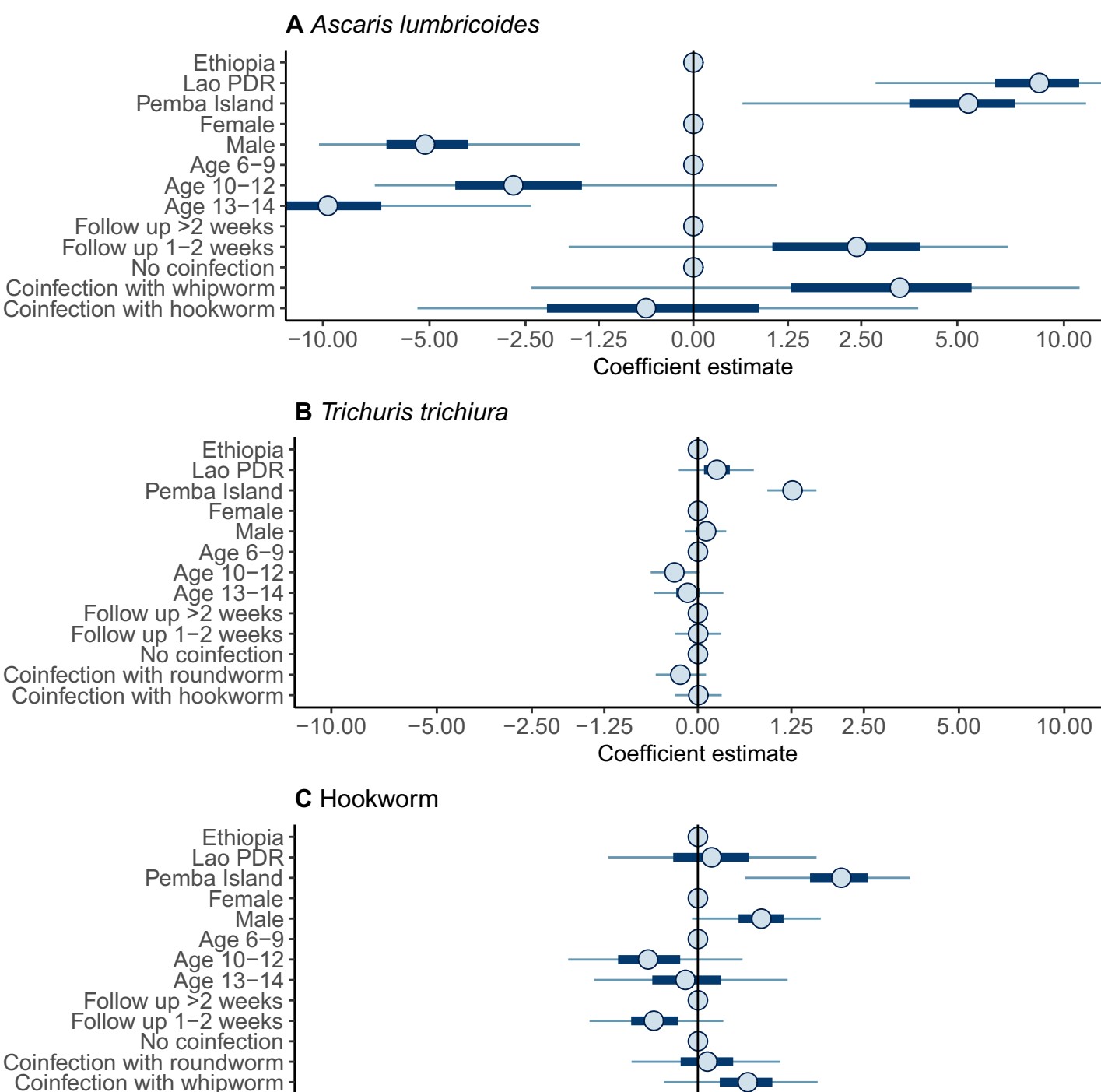

**Fig 3. Coefficients of fixed effects terms associated with the efficacy of albendazole against soil-transmitted helminth infections.** Coefficients were estimated by fitting the Bayesian model [15,16,21] to fecal egg count data on *Ascaris lumbricoides* (**panel A**), *Trichuris trichiura* (**panel B**) and hookworm (**panel C**) collected before and after administration of a single 400 mg oral dose of albendazole in three study sites, Ethiopia, Lao PDR and Pemba Island (Tanzania). Points, horizontal thick and thin lines indicate the median, the 50% and the 95% credible interval respectively. The reference category of each variable is indicated by a point on the vertical line (Ethiopia; female; age 6–9 years; follow up >2 weeks and no coinfection). Positive coefficients indicate a *lower* individual egg reduction rate, $ERR_i$. Negative coefficients indicate *a higher $ERR_i$*.

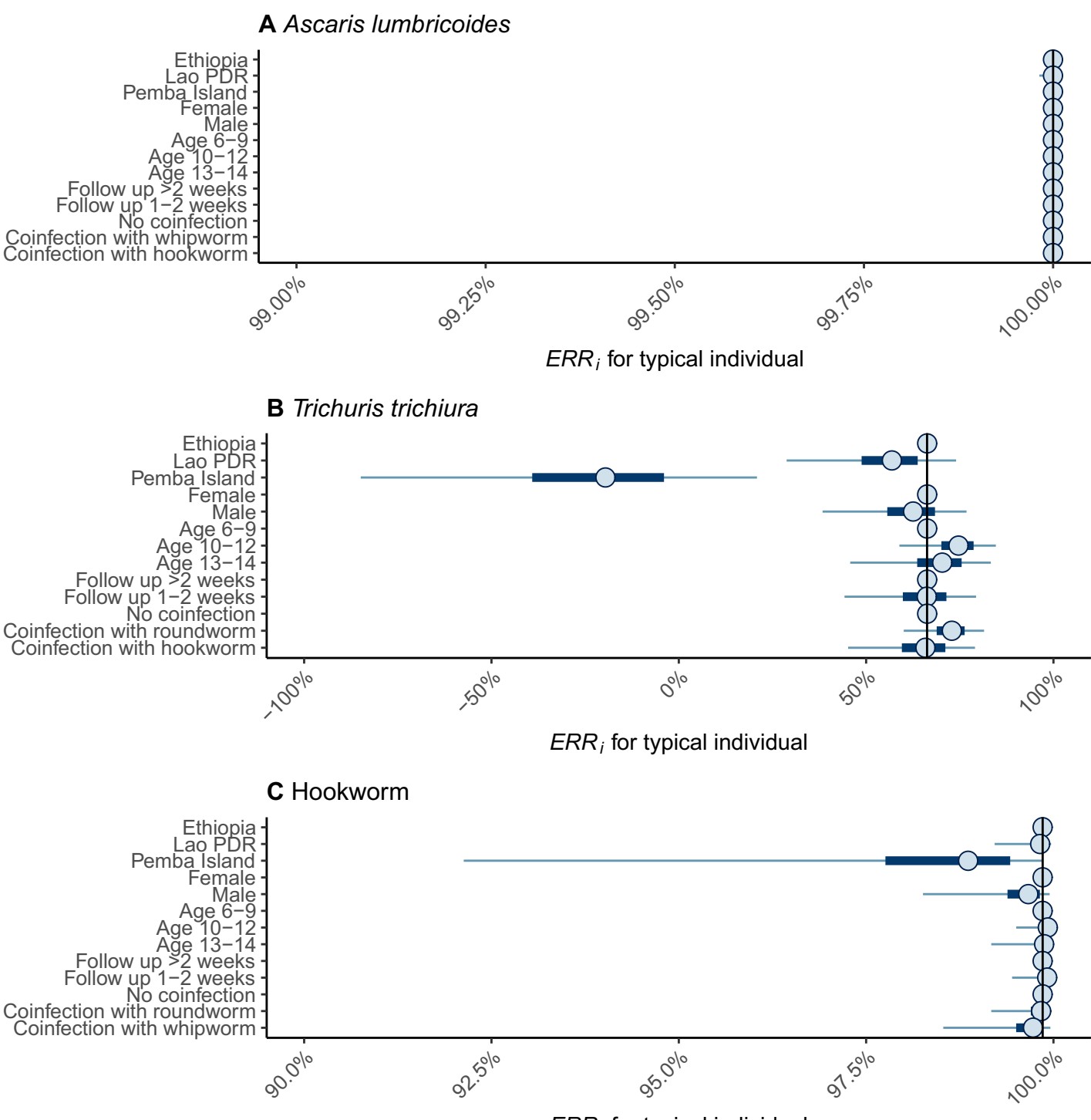

**Fig 4. Influence of fixed effects coefficients on the efficacy of albendazole against soil-transmitted helminth infections.** Individual egg reduction rates, $ERR_i$, were estimated by fitting the Bayesian model [15,16,21] to fecal egg count data on *Ascaris lumbricoides* (**panel A**), *Trichuris trichiura* (**panel B**) and hookworm (**panel C**) collected before and after administration of a single 400 mg oral dose of albendazole in three study sites, Ethiopia, Lao PDR and Pemba Island (Tanzania). Points, horizontal thick and thin lines indicate the median, the 50% and the 95% credible interval respectively. The reference $ERR_i$ (i.e., with fixed effects set to the reference categories of Ethiopia, female, age 6–9 years, follow up >2 weeks and no coinfection) is indicated by the solid vertical line. All estimates correspond to the $ERR_i$ associated with a 'typical individual' (i.e., with random effects terms set to 0).

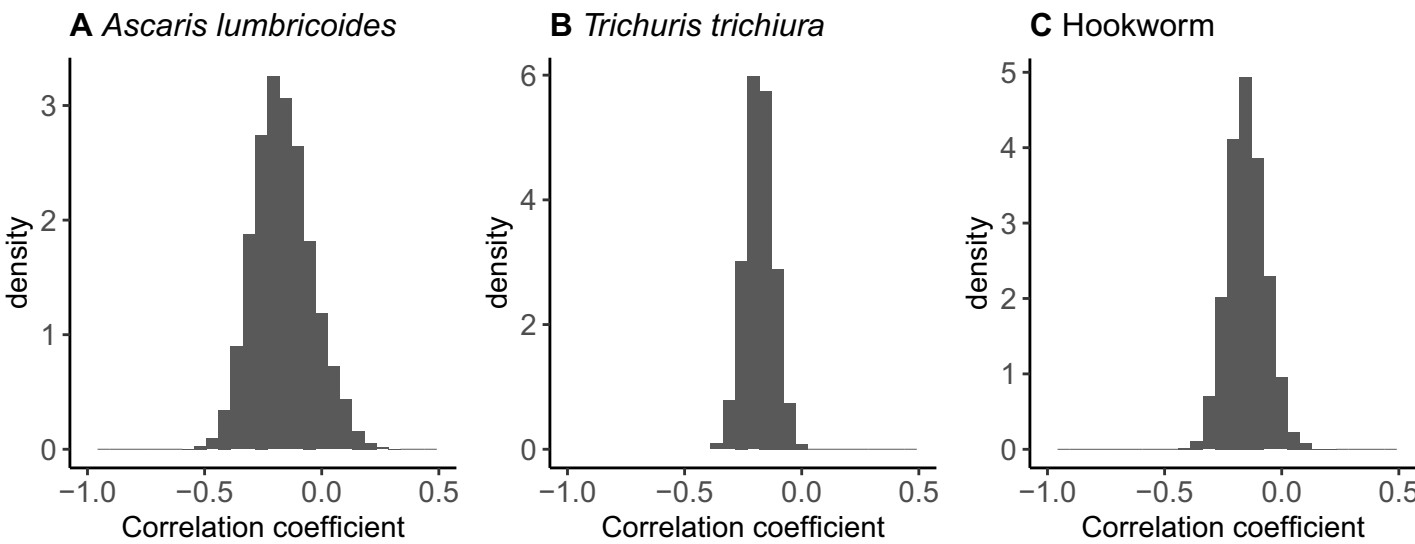

**Fig 5. Correlation between the intensity of infection before drug administration and the efficacy of albendazole against soil-transmitted helminth infections.** Correlation coefficients were estimated by fitting the Bayesian model [15,16,21] to fecal egg count data on *Ascaris lumbricoides* (**panel A**), *Trichuris trichiura* (**panel B**) and hookworm (**panel C**) collected before and after administration of a single 400 mg oral dose of albendazole in three study sites, Ethiopia, Lao PDR and Pemba Island (Tanzania).

The most worrying possible cause of the low efficacy of ALB in Pemba Island is emerging benzimidazole resistance. Yet this cannot be proven with statistical analyses of drug responses in the absence of genetic or genomic information. For example, the intensity of *T. trichiura* infection was significantly higher in the Pemba Island study participants and we found that (for all infections)—and in concordance with previous studies [11,12]—the $ERR_i$ is negatively correlated with an individual's pre-treatment FEC. Hence, although the historical drug pressure is higher in Pemba Island than in either Lao PDR or Ethiopia (since 1994 versus since 2007 and 2015 respectively)—which could make resistance more likely—we cannot distinguish the possible effects of duration of program implementation from the higher intensity of infection. It is not surprising that $ERR_i$ is negatively associated with pre-treatment intensity, since lower intensities of infection will be associated with more false negative diagnoses and thus higher apparent $ERR_i$. Density-dependent fecundity effects may exacerbate this effect in high intensity infections [31] and it is also possible that efficacy may decrease when the same drug dose is distributed among more worms. Notwithstanding, we also found that the $ERR_i$ of ALB for treating hookworm on Pemba Island was marginally lower than in the other sites, despite the pre-treatment intensity of hookworm being lower than in Lao PDR and comparable with Ethiopia.

Irrespective of the cause of the low efficacy of ALB, it is highly likely that on Pemba Island, in communities where *T. trichiura* is abundant, continuing the program with a single dose of ALB alone will not be sufficient to achieve the 2030 target of reaching and maintaining elimination of STH as a public health problem [2,32]. Elsewhere, even where benzimidazoles are of satisfactory efficacy, it is unlikely that the currently recommended monotherapy (with ALB or MEB) is an effective strategy in communities where whipworm is the dominant STH species [33]. Combination therapies using oxantel pamoate [28] or ivermectin [34,35] with ALB are more efficacious than benzimidazoles alone and our results underscore the need for co-administering drugs in areas where whipworm remains a public health problem (i.e., where infections of moderate-to-heavy intensity are still prevalent) after more than 5 years of drug. The

prequalification of a generic ivermectin of affordable price is being explored by the WHO to enable the more routine use of combination treatments where needed [36]. In addition to the improved efficacy of combination therapy, the use of more than one drug is generally a prudent approach to mitigating the likelihood of emerging resistance and safeguarding the effectiveness of control programs [37].

The key advantage of the individual-based analytical approach used here compared to more traditional population-based evaluations (e.g. [19]) is the ease of incorporating covariates that may be associated with the drug response. It also permits the effects of covariates to be distinguished from the effects that the inherent high variability in egg counts has on the $ERR_i$. For example, the population-based estimate of the ERR, $ERR_P$, for hookworm on Pemba Island indicates a substantive difference compared to the $ERR_P$ estimated in Ethiopia and Lao PDR (84% versus 96%). Yet from the individual-level analysis, it is apparent that a 'typical' participant of the Pemba Island study has only a marginally lower ERR than in Ethiopia or Lao PDR (see **Fig 4C**). This reflects the important influence that variability in egg counts has on estimates of $ERR_P$. Higher intra-participant variability in FECs (i.e., variation in FECs measured from the same individual) leads to more uncertain and left-skewed estimates of $ERR_i$. This increased uncertainty leads to the population-based $ERR_P$ being pulled substantially downwards, despite the direct effect of the covariate (Pemba Island) being rather modest.

A further advantage of individual-based analytical approaches is the ability to make individual-level statements that summarize the efficacy of the drug. From the patient perspective, this can be more useful and relevant than population-level expressions of efficacy. For example, we can state that for a (typical) child infected with hookworm, treated with ALB in Ethiopia, there was an 86% chance that the patient responded satisfactorily ($ERR_i$ >90%). Moreover, capturing individual variability in drug responses may have important consequences for population-level transmission dynamics and projections of effectiveness made using mathematical models. Typically, efficacy is considered a fixed and invariant quantity in mathematical transmission models, although variation in responses to anthelmintic drugs may deleteriously impact effectiveness of MDA programs.

Individual-level analyses of responses to anthelmintic drugs should not replace the recommended population-level methods for assessment of efficacy [17]. Population-based methods provide a straightforward means to spot check efficacy. Where suboptimal responses are suspected, individual-level methods can be used for deeper analysis to identify whether measured covariates are associated with an individual's drug response and to understand better the drivers behind population-level observations. However, a key outstanding challenge is the lack of resources for programs to monitor efficacy on a more routine basis. The vast majority of current data for assessing efficacy have been generated by research studies or clinical trials [38]. This is despite systematic monitoring by control programs being recommended by the WHO, who also offer funds for the implementation of the standard protocol. The WHO Collaborating Centre for the monitoring of anthelminthic drug efficacy for soil-transmitted helminthiasis is technically supporting this effort [17]. Notwithstanding, resistance to all major classes of anthelmintic drugs in veterinary medicine is widespread [39], perhaps a warning that more monitoring of efficacy should be undertaken during human MDA programs to safeguard continued effectiveness.

The scarcity of routine monitoring and evaluation of anthelmintic efficacy emphasizes the importance of maximizing the utility of available (primarily research) data. This is best achieved through data sharing and pooling of information, an approach perhaps best illustrated by the global collaborative effort to track the spread of antimalarial resistance [40–42]. Not only is such an approach effective for identifying and staying ahead of emerging resistance, combined analysis of large compiled datasets can provide comprehensive evaluations of

drug regimens and protocols and inform best practice to optimize treatment outcomes (e.g. [43,44]). The Infectious Diseases Data Observatory has launched a data platform for STH (and schistosomiasis, https://www.iddo.org/research-themes/schistosomiasis-sths) among other neglected tropical diseases, to facilitate sharing and reuse of individual patient data. This represents an important new resource for monitoring and optimization of anthelmintic treatment.

## Conclusion

We have re-analyzed publicly available data on individual responses to ALB from participants in three locations with differing histories of drug pressure using a powerful and contemporary statistical modelling technique. We confirm previously reported findings of treatment failure against whipworm on Pemba Island (Tanzania), estimating that only 29% of participants responded satisfactorily. Pemba Island has the highest historical drug pressure compared to the two other study sites in Ethiopia and Lao PDR, although we are unable to confirm definitively whether our findings indicate emerging benzimidazole resistance or whether they are driven by the high intensity of *T. trichiura* in Pemba Island. Current recommendations of monotherapy with benzimidazoles will be insufficient to achieve the 2030 target to achieve and maintain elimination of STH as a public health problem in Pemba Island and alternative combination therapy strategies are needed where whipworm is highly endemic.

## Supporting information

**S1 Text. Details of the Bayesian statistical model used to estimate individual egg reduction rates**
(DOCX)

**S1 Table. The estimated percentage of individuals with an a 'satisfactory' response**
(DOCX)

**S2 Table. Coefficient estimates for the negative binomial mixed effects model fitted to *Ascaris lumbricoides* fecal egg counts**
(DOCX)

**S3 Table. Coefficient estimates for the negative binomial mixed effects model fitted to *Trichuris trichiura* fecal egg counts.**
(DOCX)

**S4 Table. Coefficient estimates for the negative binomial mixed effects model fitted to hookworm fecal egg counts.**
(DOCX)

## Acknowledgments

First and foremost, the authors would like to express their gratitude towards all the children, their parents, the schoolteachers and principals who participated in this study. Second, we wish to specifically thank all the people who provided the necessary laboratory and logistic support in each of the four different sampling sites. This work would not have been possible without their willful participation and assistance.

## Author Contributions

**Conceptualization:** Martin Walker, Bruno Levecke.

**Data curation:** Piet Cools, Johnny Vlaminck, Bruno Levecke.

**Formal analysis:** Martin Walker.

**Funding acquisition:** Martin Walker, Marco Albonico, Jennifer Keiser, Jozef Vercruysse, Bruno Levecke.

**Investigation:** Martin Walker, Piet Cools, Shaali M. Ame, Mio Ayana, Daniel Dana, Leonardo F. Matoso, Simone A, Pinto, Somphou Sayasone, Johnny Vlaminck, Bruno Levecke.

**Methodology:** Martin Walker.

**Project administration:** Bruno Levecke.

**Supervision:** Piet Cools, Johnny Vlaminck, Bruno Levecke.

**Validation:** Martin Walker, Piet Cools, Johnny Vlaminck, Bruno Levecke.

**Visualization:** Martin Walker.

**Writing – original draft:** Martin Walker, Bruno Levecke.

**Writing – review & editing:** Martin Walker, Piet Cools, Marco Albonico, Shaali M. Ame, Mio Ayana, Daniel Dana, Jennifer Keiser, Leonardo F. Matoso, Antonio Montresor, Zeleke Mekonnen, Rodrigo Corrêa-Oliveira, Simone A, Pinto, Jozef Vercruysse, Johnny Vlaminck, Bruno Levecke.

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
