## [Decision Letter · Decision Letter 0]

25 Jun 2021

Dear Dr Walker,

Thank you very much for submitting your manuscript "Individual responses to a single oral dose of albendazole indicate reduced efficacy against soil-transmitted helminths in an area with high drug pressure" for consideration at PLOS Neglected Tropical Diseases. As with all papers reviewed by the journal, your manuscript was reviewed by members of the editorial board and by several independent reviewers. The reviewers appreciated the attention to an important topic. Based on the reviews, we are likely to accept this manuscript for publication, providing that you modify the manuscript according to the review recommendations. 

Sincerely,

Sabine Specht

Associate Editor

Sara Lustigman

Deputy Editor

Reviewer's Responses to Questions

**Key Review Criteria Required for Acceptance?**

**Methods**

-Are the objectives of the study clearly articulated with a clear testable hypothesis stated?

-Is the study design appropriate to address the stated objectives?

-Is the population clearly described and appropriate for the hypothesis being tested?

-Is the sample size sufficient to ensure adequate power to address the hypothesis being tested?

-Were correct statistical analysis used to support conclusions?

-Are there concerns about ethical or regulatory requirements being met?

Reviewer #1: adequately described

should add description of bootstrapping for 95%CIs

Reviewer #2: the objectives of the study Are clearly articulated with a clear testable hypothesis stated

the study design is appropriate to address the stated objectives

the population is clearly described and appropriate for the hypothesis being tested

the sample size is sufficient to ensure adequate power to address the hypothesis being tested

statistical analysis used were correct to support conclusions

there are no concerns about ethical or regulatory requirements being met

Reviewer #3: Methods paper sound

I would recommend a statistical review as the analytic methods are highly specialized

**Results**

-Does the analysis presented match the analysis plan?

-Are the results clearly and completely presented?

-Are the figures (Tables, Images) of sufficient quality for clarity?

Reviewer #1: see comments

NB: as to the question below 'Data availability' I answered 'Yes' based on the statement in the paper that the data are publicly available. However, I raised the question - see below - as to where these data are.

Reviewer #2: the analysis presented match the analysis plan

the results are clearly and completely presented

the figures (Tables, Images) are of sufficient quality for clarity?

Reviewer #3: Results are clear and informative

**Conclusions**

-Are the conclusions supported by the data presented?

-Are the limitations of analysis clearly described?

-Do the authors discuss how these data can be helpful to advance our understanding of the topic under study?

-Is public health relevance addressed?

Reviewer #1: see comments

Reviewer #2: the conclusions are supported by the data presented

the limitations of analysis are clearly described

the authors discuss how these data can be helpful to advance our understanding of the topic under study

public health relevance is addressed

Reviewer #3: Conclusions are justified by the methods used and results presented

**Editorial and Data Presentation Modifications?**

Reviewer #1: (No Response)

Reviewer #2: In the PLOS NTD figure legend tips it is stated to “avoid lenghty description of methods”. However figure and table legends describe elements of methodology not described in the methods paragraph. This could be improved.

Reviewer #3: 1. Not clear what the authors are referring to with, “..dose-response relationships against the different STH infections” (page 6, line 63)

2. The manuscript needs to be reviewed closely for typos, etc

**Summary and General Comments**

Reviewer #1: PNTD-D-21-00696

An interesting paper combining population- and individual patient-level assessment of responses to albendazole by different STHs. 

Recommend acceptance, pending some questions:

- What does “publicly available individual patient data” mean? Where are these datasets and can they be accessed by others? Refs 18, 19, 20 do not direct to raw data sources. Ae these data in the IDDO data platform? If so it should be explicitly stated from the outset. 

- Confused about the numbers. “We re-analyzed 457 individual patient data” in the Abstract does not correspond to any of the numbers reported later in the paper. Lines 144-145: “available for 645 children across the three study sites (Ethiopia: 161 cases; Lao PDR: 239 cases; Pemba Island: 245 cases” – see also next question. 

- Did this analysis include all the patients from those 3 trials? From the text and Table 1 I get the following numbers of infections. It would also be useful to have both number of infections and number of patients in the same table. Coinfections occurred. Are these a total of 1354 infections in 645 (line 145) or 441 (line 149) or 457 (Abstract) children?

 Ethiopia Lao Pemba total/species

A. lumbricoides 137 111 193 441

T. trichiura 106 105 245 456

Hookworms 90 228 139 457

total/country 333 444 577 1354

- Is it correct to assume “duplicate KK” is performed on a single stool sample?

- While understanding the rational for analyses being based on duplicate KK, was any sensitivity analysis done using another technique? This is relevant since they found “a significant correlation between the individual response and the infection intensity prior to drug administration (the individual response decreasing as a function of increasing infection intensity)”, while at the same time “egg counts that were significantly different from those reported by Kato-Katz thick smear (Mini-FLOTAC and FECPAKG2)”. 

- As hookworm infections were found to be caused by N. americanus, it may be useful to discuss different susceptibilities of N. americanus vs A. duodenalis. 

- Re: ERR. It’d be useful to have “The 95% confidence intervals (CIs) were calculated using a non-parametric block bootstrap approach [26]” in the M&Ms and not (just) in the Table 1 legend. 

- Re: WHO response classification criteria. 

o While it is customary to use ERR, whether this or rather 95%Cis should be used is still an unresolved issue.

o It is questionable to apply WHO classification criteria developed for population-level response for individual response distribution

o It is also questionable to keep using bespoke efficacy criteria that have been developed out of conveniency for poorly-effective drugs (notably: 50% ERR for ALB on T.trichiura) and continue to pretend we are happy with that. 

- I am very happy to see individual-level response finally being adopted to assess the level of efficacy of anthelminthics.

Reviewer #2: This a very interesting and beautiful secondary application of bayesian count models to estimate effect of patient’s and study characteristics on treatment efficacy on egg reduction rates from 3 pooled study data. The text is well writtten despite the complexity of the subject.

There are some minor comments to slightly improve the manuscript.

Page 9, FEC: the authors should mention how FEC is obtained as it is stated later at several occasion especially in the legend of figure 1, first sentence after figure 1 and legend of table 2

Page 12, table 1 legend: the authors define the number of cases as the number of participants identified with any four diagnostics. However they have previously stated in the methods that the ERR is calculated only with the KK evaluation. Therefore they should explain how is calculated the ERR in the absence of KK evaluation and with one of the 3 other diagnostic tool.

Reviewer #3: (No Response)

PLOS authors have the option to publish the peer review history of their article (what does this mean?). If published, this will include your full peer review and any attached files.

Reviewer #1: Yes: Prof Piero Olliaro

Reviewer #2: Yes: Michel Vaillant

Reviewer #3: No

Figure Files:

Data Requirements:

Reproducibility:

References

---

## [Decision Letter · Decision Letter 1]

8 Oct 2021

Dear Dr Walker,

We are pleased to inform you that your manuscript 'Individual responses to a single oral dose of albendazole indicate reduced efficacy against soil-transmitted helminths in an area with high drug pressure' has been provisionally accepted for publication in PLOS Neglected Tropical Diseases.

Best regards,

Sabine Specht

Associate Editor

Sara Lustigman

Deputy Editor

Reviewer's Responses to Questions

**Key Review Criteria Required for Acceptance?**

**Methods**

-Are the objectives of the study clearly articulated with a clear testable hypothesis stated?

-Is the study design appropriate to address the stated objectives?

-Is the population clearly described and appropriate for the hypothesis being tested?

-Is the sample size sufficient to ensure adequate power to address the hypothesis being tested?

-Were correct statistical analysis used to support conclusions?

-Are there concerns about ethical or regulatory requirements being met?

Reviewer #1: well described

Reviewer #2: the objectives of the study Are clearly articulated with a clear testable hypothesis stated

the study design is appropriate to address the stated objectives

the population is clearly described and appropriate for the hypothesis being tested

the sample size is sufficient to ensure adequate power to address the hypothesis being tested

statistical analysis used were correct to support conclusions

there are no concerns about ethical or regulatory requirements being met

Reviewer #3: Methods are well described and appropriate. No additional comments.

**Results**

-Does the analysis presented match the analysis plan?

-Are the results clearly and completely presented?

-Are the figures (Tables, Images) of sufficient quality for clarity?

Reviewer #1: well presented

Reviewer #2: the analysis presented match the analysis plan

the results are clearly and completely presented

the figures (Tables, Images) are of sufficient quality for clarity

Reviewer #3: Results are clearly presented. No additional comments.

**Conclusions**

-Are the conclusions supported by the data presented?

-Are the limitations of analysis clearly described?

-Do the authors discuss how these data can be helpful to advance our understanding of the topic under study?

-Is public health relevance addressed?

Reviewer #1: well done, thorough

Reviewer #2: the conclusions are supported by the data presented

the limitations of analysis are clearly described

the authors discuss how these data can be helpful to advance our understanding of the topic under study

public health relevance is addressed

Reviewer #3: Conclusions are supported by data presented

**Editorial and Data Presentation Modifications?**

Reviewer #1: none

Reviewer #2: no specific comment

Reviewer #3: (No Response)

**Summary and General Comments**

Reviewer #1: the authors have satisfactorily addressed the comments made on the first version. Recommend approval

Reviewer #2: no specific comment

Reviewer #3: Important and clearly presented data. Limitations are adequately discussed.

PLOS authors have the option to publish the peer review history of their article (what does this mean?). If published, this will include your full peer review and any attached files.

Reviewer #1: **Yes: **Prof Piero L Olliaro

Reviewer #2: **Yes: **Michel Vaillant

Reviewer #3: No

---

## [Editor Report · Acceptance letter]

15 Oct 2021

Dear Dr Walker,

We are delighted to inform you that your manuscript, "Individual responses to a single oral dose of albendazole indicate reduced efficacy against soil-transmitted helminths in an area with high drug pressure," has been formally accepted for publication in PLOS Neglected Tropical Diseases.

Best regards,

Shaden Kamhawi

co-Editor-in-Chief

Paul Brindley

co-Editor-in-Chief
